# Target DNA bending by the Mu transpososome promotes careful transposition and prevents its reversal

**James R Fuller, Phoebe A Rice***

Department of Biochemistry and Molecular Biology, The University of Chicago, Chicago, United States

**Abstract** The transposition of bacteriophage Mu serves as a model system for understanding DDE transposases and integrases. All available structures of these enzymes at the end of the transposition reaction, including Mu, exhibit significant bends in the transposition target site DNA. Here we use Mu to investigate the ramifications of target DNA bending on the transposition reaction. Enhancing the flexibility of the target DNA or prebending it increases its affinity for transpososomes by over an order of magnitude and increases the overall reaction rate. This and FRET confirm that flexibility is interrogated early during the interaction between the transposase and a potential target site, which may be how other DNA binding proteins can steer selection of advantageous target sites. We also find that the conformation of the target DNA after strand transfer is involved in preventing accidental catalysis of the reverse reaction, as conditions that destabilize this conformation also trigger reversal.

## Introduction

***For correspondence:** price@ uchicago.edu

**Competing interests:** The authors declare that no competing interests exist.

Transposons are mobile DNA elements that move or copy their DNA sequence from one location to another. They have exhibited a remarkable ability to spread, such that sequences derived from transposons are pervasive in the genomes of prokaryotes and eukaryotes alike (*Aziz et al., 2010*). Among transposons whose behavior has been examined in vitro, the transposable *Escherichia coli* bacteriophage Mu is one of the most active (*Harshey and Bukhari, 1981*; *Mizuuchi, 1983*) and well-studied (reviewed in *Harshey, 2012*). Its transposase, MuA, belongs to the large DDE family of recombinases (*Baker and Luo, 1994*; *Rice and Mizuuchi, 1995*). This family is named for the amino acid residues in their shared RNase-H-like catalytic domains that bind the divalent metals necessary for catalysis. In addition to the transposases for many common transposons, the DDE family also includes retroviral integrases, which use the same reaction mechanism to integrate viral genomes into host chromatin (*Fujiwara and Mizuuchi, 1988*; *Li et al., 2006*).

To catalyze transposition (*Figure 1A*), DDE recombinases like MuA bind specific sequences at each end of their element and synapse them together in a complex known as the transpososome (or, for retroviral integrases, the intasome) (*Surette et al., 1987*; *Wei et al., 1997*). The transpososome then hydrolyzes the phosphate backbone at the boundary between the transposon and flanking host DNA, and catalyzes the attack of the resulting 3' hydroxyl groups from each transposon end into the 'target' destination DNA. This critical second chemical step is referred to as strand transfer. In the case of bacteriophage Mu, the entire prophage acts as a transposon, and strand transfer occurs at positions 5 bp apart in the target DNA (*Grindley and Sherratt, 1979*). Rather than catalyzing multiple turnovers like a typical enzyme, MuA remains bound to the branched strand transfer products until forcibly disassembled by the ATP-powered host chaperone ClpX (*Burton et al., 2001*; *Mhammedi-Alaoui et al., 1994*), after which transposition can be completed

**eLife digest** Pieces of DNA called transposons can move or copy themselves around the genome. Some viruses – such as HIV and Mu (a virus that infects bacteria) – act as transposons to hide their DNA by inserting it into their host's genome.

Mu, HIV and many transposons all work in the same, somewhat unusual way. Like many chemical reactions, joining DNAs together needs a source of energy to make it happen, yet these viruses and transposons do not need high energy inputs to work. In addition, they do not look for a specific DNA sequence to insert their DNA into. This gives them the advantage of inserting copies of their DNA anywhere in the host's genome, but also means that multiple copies might mistakenly insert into each other.

Visualizations of the insertion process show that the DNA that the viruses insert their DNA into is always bent like a U-turn. Why does this bending occur? It may be that the bending helps the virus to choose where in the DNA to insert and acts as a way to power the chemical reaction that joins the DNA. To investigate this possibility, Fuller and Rice performed experiments using purified fragments of DNA and the enzyme from Mu that does the DNA joining chemistry. The results revealed that making the insertion site DNA easier to bend made the insertion much faster. Furthermore, a mutant enzyme that struggled to bend the DNA had trouble keeping the chemistry going, and so the viral DNA would accidentally pop back out after it was joined. Thus the insertion site DNA is like a spring: the enzyme puts a lot of energy into bending it, but once the viral DNA has been inserted that energy is released to power the reaction to completion.

Fuller and Rice conclude that if other proteins were to pre-bend or otherwise make the DNA more flexible, this would tell the DNA-joining enzyme where to insert, which helps explain the roles of known targeting proteins for Mu and HIV. Further work is now needed to investigate whether these other targeting proteins exist for other viruses and transposons, and to identify them.

by the host's own DNA repair and replication machinery (*Mizuuchi, 1984*; *North and Nakai, 2005*). Although MuA is still active after removal by ClpX in vitro, it may be degraded by the ClpXP chaperone-protease complex in vivo (*Levchenko et al., 1995*).

Transposases and integrases face two challenges when interacting with target DNA sites. First, successful transposition requires that they avoid selecting their own DNA as a target, as intra-self strand transfer can lead to deletion of some or all of the mobile element and a double strand break. To this end, MuA and some other DDE transposases interact with a partner protein whose role is to remain bound only to distant, non-self target sites. For Mu, this is the MuB protein encoded by the phage itself (*Maxwell et al., 1987*; *Mizuno et al., 2013*), whereas retroviral integrases bind to host nucleosomes to choose a target (*Pryciak and Varmus, 1992*). However, the mechanism(s) by which transpososomes resist the high local concentration of self-DNA but are activated to attack DNA bound by their target selection partner are not well understood.

After strand transfer, the transposase must also avoid catalysis of the reverse reaction (termed 'disintegration') in the time before host machinery completes transposition. Strand transfer merely exchanges of one pair of 3' hydroxyl groups and phosphodiester bonds for another and so should not provide a net release of chemical bond energy that could drive the reaction towards its products. Nevertheless, the Mu transpososome displays a strong bias towards catalysis of only the forward strand transfer reaction (*Au et al., 2004*; *Lemberg et al., 2007*; *Mizuuchi et al., 2007*). It seems likely that this bias stems in some way from product binding energy, as transpososomes remain tightly bound to their products.

Here we show that target DNA bending contributes to overcoming both of the above challenges. Four crystal structures of transpososomes and intasomes from the DDE family that include the target DNA are available, including that of the Mu transpososome (*Figure 1C*) (*Maertens et al., 2010*; *Montaño et al., 2012*; *Morris et al., 2016*; *Yin et al., 2016*). Beyond the shared catalytic domain fold, the specifics of the protein-protein and protein-DNA contacts in these structures are very different. Nevertheless, they have all converged on a bend in the target DNA following strand transfer. Clues that target DNA bending can play a role in target site selection come from studies of target

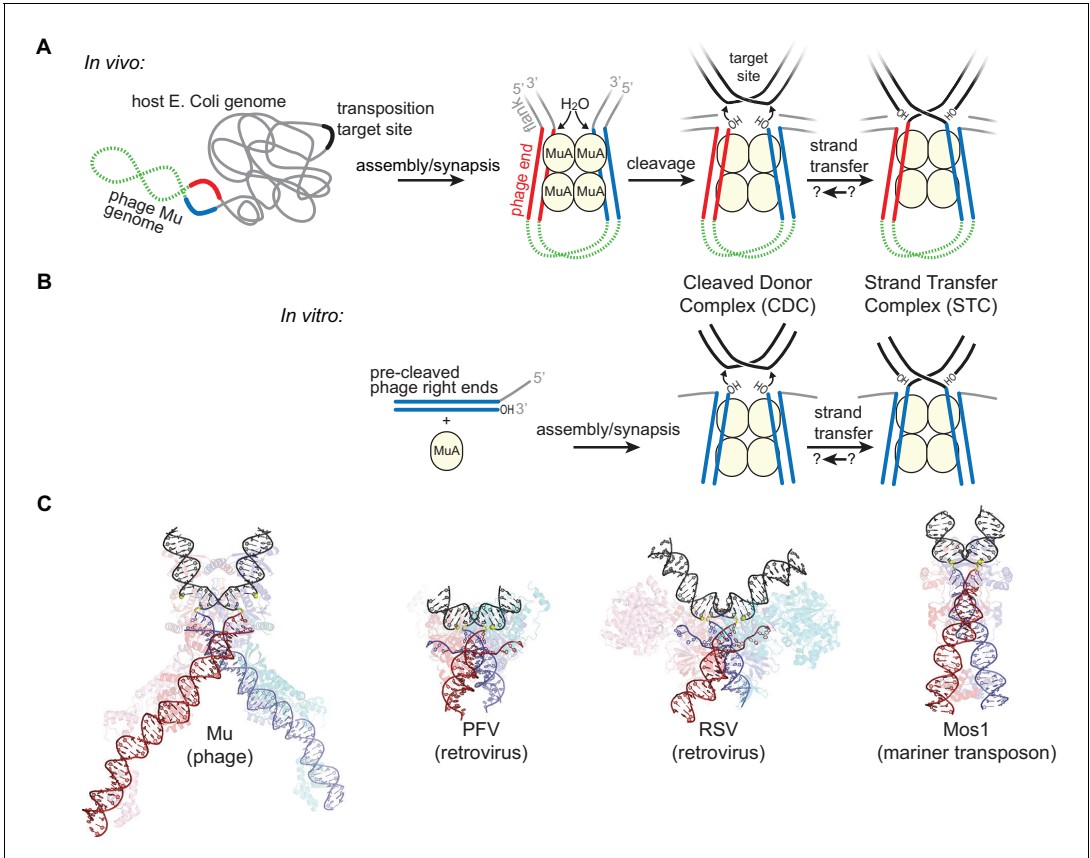

**Figure 1.** Transposition by Bacteriophage Mu and Available Structures of DDE family members. (**A**) Diagram of replicative transposition. The transposable element (here, the phage genome) is in green with transposase binding sites in red and blue. Transposase subunits (here, MuA, in light yellow circles) synapse element ends and catalyze their nicking and subsequent joining to target DNA (black). In vivo, the flanking DNA (grey) is double stranded and may represent the entire host chromosome, which may also provide the target site. (**B**) The in vitro Mu transposition system used here utilizes short linear fragments for both the target DNA and phage ends, with 3 nt of single stranded flanking DNA. (**C**) Strand transfer complex structures. DNA is colored as in (**A**), with protein components partially transparent and colored according to their bound DNA. Generated from PDB IDs 4FCY (Mu), 3OS0 (PFV), 5HOO (Mos1), and 5EJK (RSV).

The following figure supplement is available for figure 1:

**Figure supplement 1.** Mu end and target DNA fragments.

site preferences. In the absence of target selection partners, Mu and retroviral integrases prefer target sites with more easily deformable sequence steps, suggesting that DNA flexibility is interrogated prior to strand transfer and can guide target site selection (*Haapa-Paananen et al., 2002*; *Pryciak and Varmus, 1992*; *Serrao et al., 2015*). This is supported by a structure of the prototype foamy virus (PFV) intasome bound to, but not integrated into, target DNA, which shows the target DNA bent in a conformation very similar to that in the strand transfer complex (*Maertens et al., 2010*). On the other hand, it has also been suggested that target DNA bending could prevent the reversal of strand transfer by driving the products (the new target 3' hydroxyl group and transposon-target phosphodiester bond) out of the active sites to reduce the conformational strain of the bend (*Maertens et al., 2010*; *Montaño et al., 2012*). Both ideas have yet to be tested directly in vitro.

In this work, we measure how target DNA flexibility and bending affect target DNA binding, strand transfer, and disintegration by the Mu transpososome. We show that increasing DNA flexibility has a dramatic positive effect on the transposition reaction and that bending occurs during binding, implying that bending is required but carries a steep energetic cost. Afterwards, disintegration

is very rare and occurs under extreme conditions that coincide with disruption of the bend. Further, a mutant transpososome with compromised target DNA binding and bending can be rescued by pre-bent DNA and is particularly prone to strand transfer reversal. Our results are the first to biochemically link unbending to reversal, and also point to target DNA bending as a key energetic barrier to strand transfer. This barrier would allow DNA deformation generated by other proteins to steer target site selection, and provide a way to channel product binding energy into preventing strand transfer reversal.

## Results

### Flexible or bent DNA is a highly reactive transposition target

Mu is an attractive model system because transpososomes can be assembled in vitro from MuA protein and short DNA substrates (*Figure 1B*) (*Savilahti et al., 1995*). The DNAs we use here mimic the products of transposon-host junction cleavage, thus removing that earlier reaction step from our analysis (compare *Figure 1A* versus *Figure 1B*). They retain only 3 nt of flanking DNA on the non-transferred strand (*Figure 1—figure supplement 1*).The assembled transpososome without target DNA is referred to as the cleaved-donor complex (CDC), and a transpososome that has bound and attacked target DNA as the strand transfer complex (STC). All MuA constructs used here lacked domain Iα (residues 1–76), a site-specific DNA binding domain which in vivo binds an enhancer sequence within the interior of the Mu genome to aid transpososome assembly (*Mizuuchi and Mizuuchi, 1989*). This domain was also omitted in the crystal structure of the Mu STC (*Montaño et al., 2012*), is not required for transposition in vitro, and is in fact slightly inhibitory in the absence of the enhancer (*Yang et al., 1995*).

We first sought to determine whether target DNA flexibility has an effect on target DNA binding and the kinetics of the strand transfer reaction. We use two methods to increase the flexibility of duplex DNA: DMSO added to the reaction buffer (*Escara and Hutton, 1980*; *Herrera and Chaires, 1989*), and/or a single G:G base-pairing mismatch incorporated at the center of the target DNA sequence (*Rossetti et al., 2015*). In addition to altering the biophysical properties of DNA, both have been used in previous studies as general enhancers of the transposition activity of MuA in vitro (*Baker and Mizuuchi, 1992*; *Savilahti et al., 1995*; *Yanagihara and Mizuuchi, 2002*). Because DMSO also affects earlier transpososome assembly steps, and to eliminate spare Mu ends or MuA subunits that could compete with our intended target DNAs, we first purified the CDC form of the transpososome by gel filtration chromatography. To prevent premature catalysis of strand transfer, CDCs were assembled and purified in a buffer containing EDTA and lacking $Mg^{2+}$. A 2:1 mixture of MuA protein to Mu end DNA in this buffer results in a gel filtration peak shifted to a very high molecular weight that we identify as CDCs based on size and near-stoichiometric reactivity with target DNA (*Figure 2*).

We monitored the effect of DNA flexibility on the kinetics of strand transfer by sampling reactions containing 100 nM each of purified CDC and $^{32}$P-labeled target DNA as a function of time. In these experiments, the target DNA strand is labeled at the 5′ end and becomes cleaved as a result of strand transfer. As has been observed previously (*Yanagihara and Mizuuchi, 2002*), a mismatched base pair directs the vast majority of strand transfer events to occur centered around it, hence the single dominant product (*Figure 2B*). The fully base paired target DNA, which is identical in sequence except for the central nucleotide in one strand, results in two major products (and a number of minor products, see *Figure 2—figure supplement 1*). It is likely that the major insertion site remains approximately centered even without the mismatched base pair because our 35 bp target DNA is not much larger than the total target DNA binding surface of the CDC.

Increased target DNA flexibility via the G:G mismatch or DMSO significantly enhances the rate of strand transfer (*Figure 2B,C*), without perturbing the CDCs' ability to perform concerted strand transfer with both ends (*Figure 2—figure supplement 2*). Using the steepest slope between any two timepoints in *Figure 2C* as an estimate of initial rate, the initial rate for the unmodified reaction was about 0.1 $\mu M^{-1}$ $min^{-1}$, and it required about 40 min to convert 30% of the target DNA into strand transfer product. When the reaction buffer was supplemented with 15% (v/v) DMSO, the initial rate was about 0.8 $\mu M^{-1}$ $min^{-1}$ and was 30% complete in only 4 min. The mismatch was even more powerful, leading to an initial rate of about 3.0 $\mu M^{-1}$ $min^{-1}$ and 30% completion in about 1

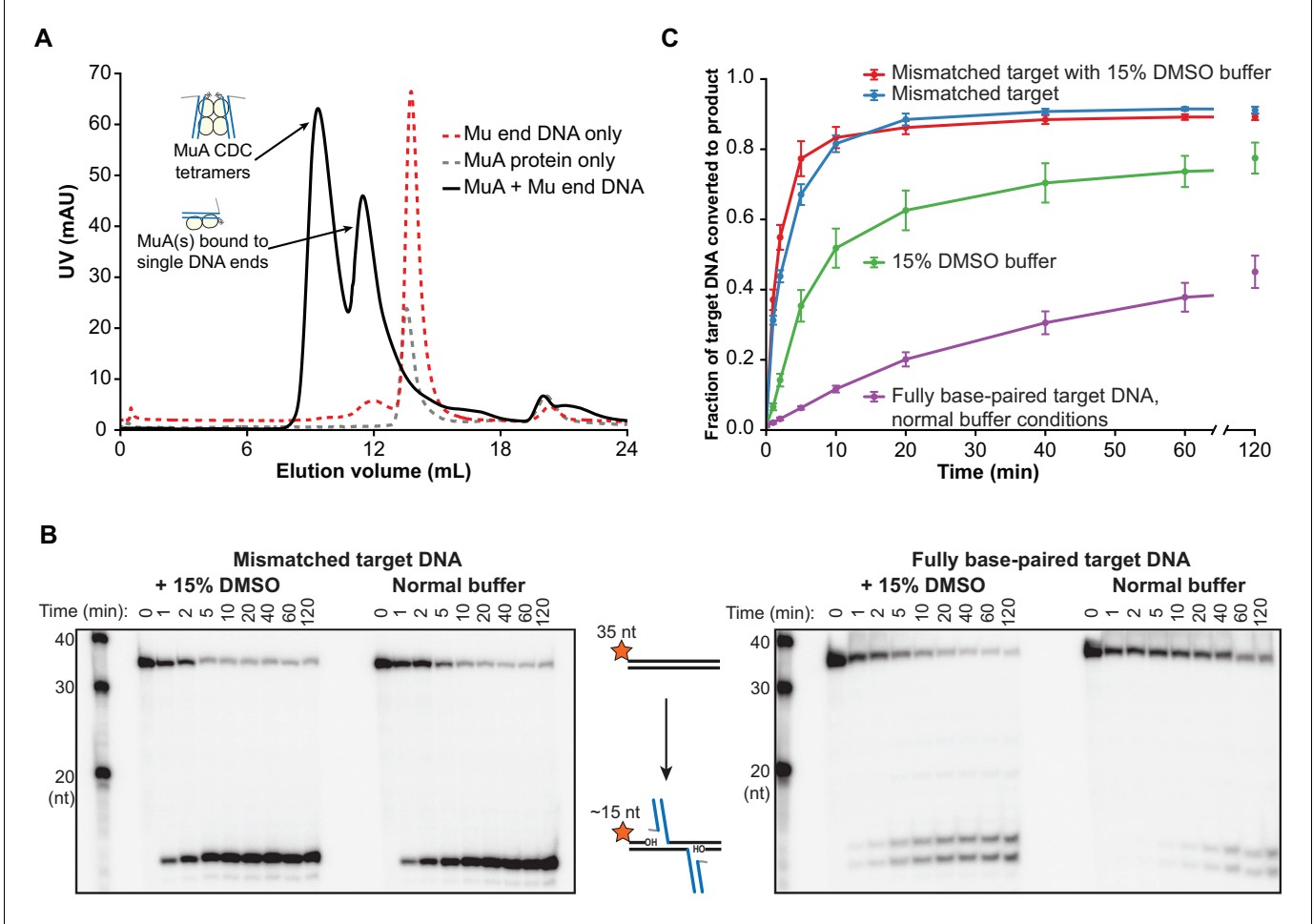

**Figure 2.** Target DNA bending during Mu transposition affects strand transfer kinetics. (**A**) Purification of transpososome tetramers constructed in vitro by gel filtration. Full transpososomes (black line) are separable from lower molecular weight complexes Also shown are MuA protein and phage end DNAs alone (dashed lines). The contents of only the tetramer peak were used in our assays here. (**B**) Strand transfer kinetics visualized by denaturing gel electrophoresis and 5'-$^{32}$P-labeled target DNA. Where indicated, the 35 bp target DNA sequence includes a G:G base-pairing mismatch and/or the reaction buffer was supplemented with 15% DMSO. Strand transfer results in cleavage of the labeled strand. (**C**) Quantification of the strand transfer kinetics experiments described in (**B**). The vertical axis represents the fraction of total lane signal present in product band(s). The X-axis is broken between 60 and 120 min to enhance the readability of early timepoints. Error bars represent mean ± standard error of the mean (SEM), n = 4 independent CDC preparations, one of which is shown in (**B**).

The following figure supplements are available for figure 2:

**Figure supplement 1.** Minor strand transfer products viewed at high contrast.

**Figure supplement 2.** Modifying target DNA flexibility or removing domain III do not alter the concerted nature of MuA-catalyzed strand transfer.

min. The combined effect of both modifications was additive, albeit only marginally (shown most clearly in the 2 and 5 min timepoints in *Figure 2C*).

To determine whether accelerated strand transfer rates result from tighter binding to the more flexible target DNA substrates, we used fluorescence anisotropy to measure the affinity of the interaction between purified CDCs and target DNA (*Figure 3A*). To measure these values under conditions where the MuA active site and the DNA ionic environment would be as realistic as possible, we performed these experiments in the presence of Mg$^{2+}$. Rather than by withholding divalent metals, we prevented strand transfer by using Mu end DNAs lacking the terminal 3'OH group on the transferred strand, which is the nucleophile for strand transfer (*Figure 3—figure supplement 1*). Binding

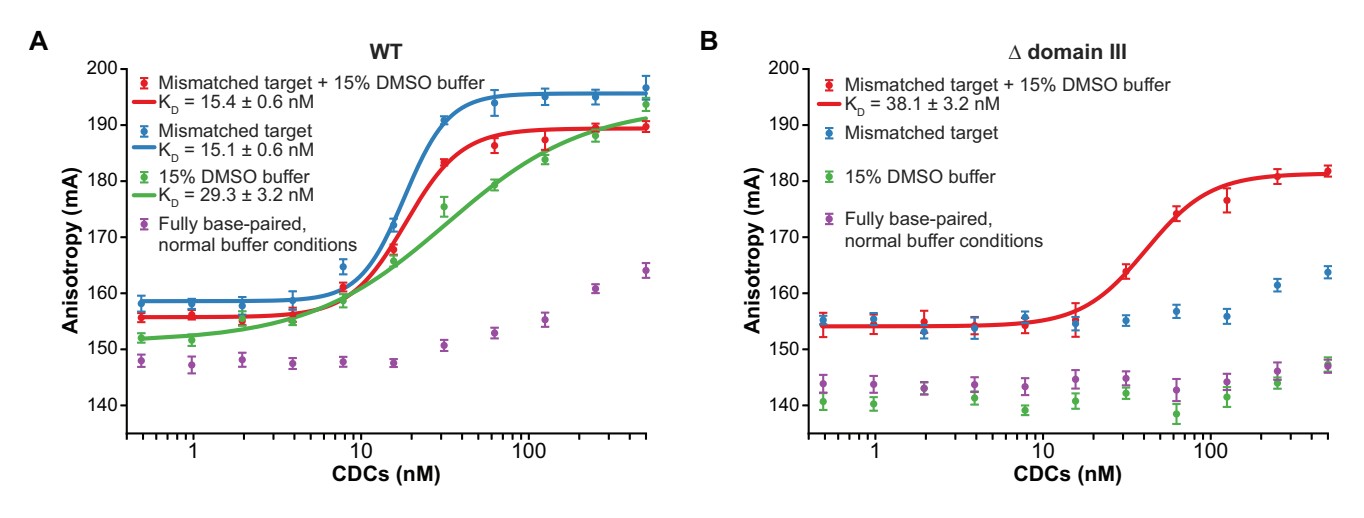

**Figure 3.** Enhanced flexibility triggers tight target DNA binding. Fluorescence anisotropy measurements of transpososome binding to Atto565-labeled target DNAs. Dots are experimental data, with error bars representing 95% confidence intervals derived from 15 measurements (see Materials and methods). Solid lines represent fits to the data to obtain the $K_D$ values indicated in the legends. (A) Wild-type transpososomes, (B) $\Delta$ domain III transpososomes (see *Figure 4A*).

The following figure supplement is available for figure 3:

**Figure supplement 1.** Strand transfer and/or target DNA nicking are blocked by terminating the transferred strand with dideoxy-adenosine.

affinities followed a similar pattern to reaction rates: The $K_D$ of CDCs for mismatched target DNA (regardless of DMSO) was about 15 nM, and for fully base paired target DNA with DMSO, about 29 nM. In the case of the mismatched target, good fits to the data required a cooperative binding model with a Hill coefficient of 3.3 (for the mismatch alone) and 2.6 (when combined with DMSO). We suspect this arises from transient CDC disassembly events that become significant at low nanomolar concentrations. Conversely, we were unable to detect enough binding to normal DNA under normal buffer conditions to confidently fit a binding curve, but can visually estimate that $K_D$ might lie between 0.5–1 µM. Thus, increasing the flexibility of the target DNA can enhance its affinity for transpososomes by at least 33-fold.

These results indicate that bending or otherwise deforming the target DNA poses a significant energetic barrier to binding target DNA. We also note that the binding and kinetic data do not exactly correspond. For instance, although the mismatch and DMSO produced similar enhancements in target DNA affinity (with $K_D$s differing by only two-fold), the mismatch had a much stronger effect on the strand transfer reaction than DMSO (with initial rates differing by about four-fold), even though the kinetic experiments were carried out at 100 nM, above the $K_D$s for either of these target interactions. This suggests that there may be additional localized target DNA conformational changes, beyond those that are required for initial binding, that are necessary to properly position the target DNA phosphate backbone in the active site, and that are better facilitated by the mismatch than by DMSO.

## MuA domain III participates in target DNA binding and bending

The high binding affinity of CDCs for flexible target DNA suggests that it must be bent in order to make optimal contacts with the transpososome, and the crystal structure of the Mu STC suggests that domain IIIα, which forms a positively charged alpha helix, stabilizes the bent target DNA conformation. The two copies of domain IIIα from the catalytic MuA subunits pair in the middle of the target DNA U-turn (*Figure 4A*), presumably countering the electrostatic repulsion between the two arms of the target DNA, as well as providing additional protein-DNA contacts. To test the importance of domain III in DNA binding, we repeated the above kinetic and binding experiments using CDCs lacking domain III on the catalytic MuA subunits. Note that domain IIIβ, which comprises the

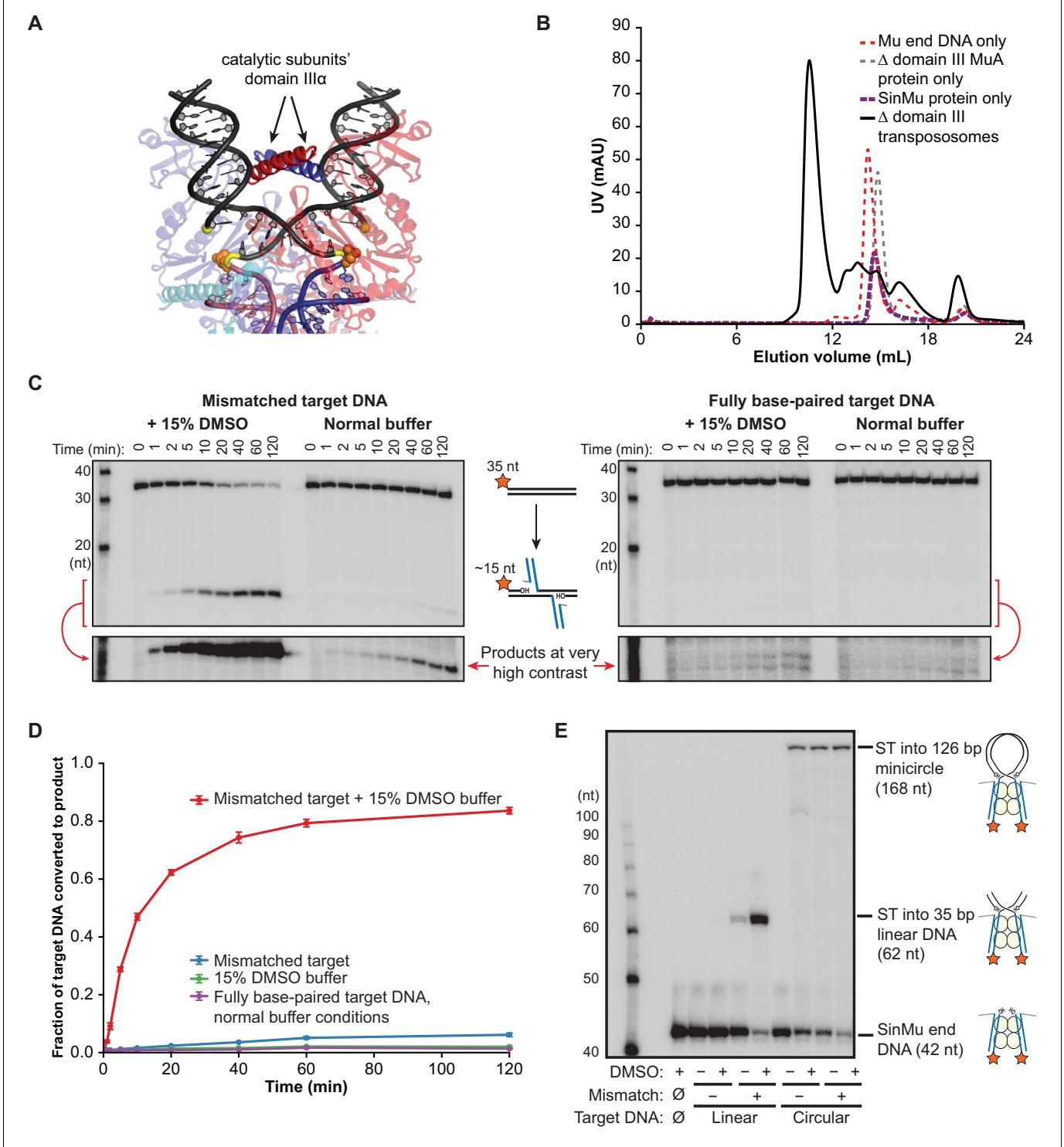

**Figure 4.** *Δ* Domain III Transpososomes are rescued by enhanced target DNA bending. (**A**) Domain III from the catalytic MuA subunits is used to make contacts to the bent target DNA. The solid (non-transparent) alpha helices indicated with arrows are the portions domain III resolved in the MuA STC crystal structure. Colors as in *Figure 1C*, with MuA active site residues as orange spheres. (**B**) Gel filtration chromatography of *Δ* domain III CDCs. Moving to the chimeric SinMu system and truncating the catalytic MuA subunits to remove domain III does not hinder formation and subsequent purification of CDCs. (**C**) Strand transfer kinetics visualized by denaturing gel electrophoresis and 5'-$^{32}$P-labeled target DNA, as in *Figure 1D* except using *Δ* domain III transpososomes. Lower panels are the product band(s) from the upper panels at greatly increased contrast. (**D**) Quantification of the strand transfer kinetics experiments described in (**C**). The vertical axis represents the fraction of total lane signal present in product band(s). Error bars

*Figure 4 continued on next page*

*Figure 4 continued*

represent mean ± SEM, n = 4 CDC preparations, one of which is shown in (C). (E) Rescue of the strand transfer activity of truncated transpososomes by circular DNAs. $^{32}$P-labeled Mu ends increase in size as a result of strand transfer. Samples were taken 2 hr after transpososomes were mixed with Mg$^{2+}$ and indicated target DNAs.

The following figure supplements are available for figure 4:

**Figure supplement 1.** The SinMu system does not inherently alter interactions with target DNA.

**Figure supplement 2.** Minicircle target DNA.

**Figure supplement 3.** Minicircle rescue of Δ domain III transpososome strand transfer activity is not an artifact of the sequences of the DNAs used to form the minicircles.

C-terminal ~60 residues of MuA, was not included in the crystal structure because is not required for transposition in vitro, but has been included in the constructs used in this work. Although domain III can be removed by truncating MuA constructs at residue 560, transpososome assembly requires domain IIIα to be present on the two MuA subunits not involved in catalysis (*Aldaz et al., 1996*). In order to localize truncated subunits only at the catalytic positions, we utilized our 'SinMu' system (*Figure 4—figure supplement 1*) (see also *Ling et al., 2015*). SinMu is a chimeric protein in which the sequence-specific DNA binding domains of MuA have been replaced with that of the unrelated Sin recombinase. It can be placed at the non-catalytic positions in the transpososome by a corresponding substitution in the Mu end DNA sequence (*Figure 1—figure supplement 1*). Chimeric 'SinMu'-containing CDCs that include the full MuA C-terminus at all subunit positions behave indistinguishably from wild type CDCs (*Figure 4—figure supplement 1*), and subsequent truncation of the catalytic subunits' domain III does not hinder assembly and purification (*Figure 4B*).

We found that CDCs where the catalytic MuA subunits lack domain III ('Δ domain III CDCs') display very little strand transfer activity, except under particular conditions. Rather than just enhancing the reaction, the combination of DMSO and a target base-pairing mismatch were nearly essential for the truncated CDCs to catalyze strand transfer into short linear target DNAs at detectable levels (*Figure 4C,D*). The reaction rate is otherwise exceedingly slow (<1% of input after two hours under the next best condition, the mismatch alone). Measurements of target DNA binding by the truncated CDCs once again followed a similar pattern to strand transfer kinetics: binding was relatively robust when both modifications are present simultaneously ($K_D \approx 38$ nM), but undetectable otherwise (*Figure 3B*). Thus, domain III does indeed provide many of the protein-DNA contacts that are important to capture a target. High target DNA flexibility, however, can make up for the loss of these contacts. This confirms our assertion that the contacts are optimized for bent DNA.

To investigate further the connection between binding and bending, we tested whether the strand transfer activity of the truncated CDCs could be rescued by providing pre-bent DNA for use as a target, rather than the linear DNA fragments used thus far. We generated pre-bent DNA by creating 126 bp DNA minicircles (*Figure 4—figure supplement 2*). Instead of just being more flexible, minicircles should be naturally and permanently held in a bent state. This should provide an even lower energy barrier to bending. Accordingly, we found that minicircle DNAs can rescue the strand transfer activity of Δ domain III CDCs under a number of conditions (*Figure 4E* and *Figure 4—figure supplement 3*). While Δ domain III CDCs generate a substantial amount of strand transfer products into linear DNA only in the presence of both the mismatched base-pair and DMSO, strand transfer into minicircles requires only one or the other. Thus the loss of transpososome-target DNA interactions can be mitigated by pre-bending the DNA. This is also further evidence that DMSO and the base-pairing mismatch allow the DNA to be more easily bent, and that target DNA binding is linked to bending.

## Strand transfer reversal only occurs under conditions that would compromise target DNA binding

The strand transfer reaction does not change the overall number or type of covalent bonds, and is entropically unfavorable in that it links together previously separate DNA segments. The free energy

landscape of this reaction might thus be expected to favor its reversal (hereafter referred to as 'disintegration'). Nevertheless, our data above show that strand transfer is capable of going nearly to completion (*Figure 2C*), indicating that the STC is able to suppress disintegration. Given that Mu transpososomes are not true catalysts (each transpososome catalyzes a single strand transfer reaction, and then remains bound to the reaction products), it is likely that MuA suppresses disintegration by using product binding energy to bias the reaction direction toward the final product complex. Our finding that CDCs bind particularly tightly to artificially flexible DNA, and the observation that strand transfer should make any target DNA more flexible by nicking both strands, led us to investigate the relationship between target DNA conformation and disintegration.

We first wanted to establish the baseline rate at which the STC catalyzes disintegration. Published reports currently differ regarding this rate. It is difficult to address because it requires a starting pool of pure STCs (free of unreacted target DNA, which would be indistinguishable from the product of disintegration), and any purification protocols used must preserve the activity and structure of the STCs. This has been previously approached, to our knowledge, in two ways: (1) forming STCs by mixing MuA, Mu end DNAs and target DNA and $Mg^{2+}$, as was done for crystallization (*Montaño et al., 2012*), then purifying the resulting STCs by electrophoresis in an agarose gel in TBE buffer (which chelates $Mg^{2+}$), and (2) circumventing the issue of unreacted target DNA by directly assembling transpososomes on branched DNA substrates that mimic the strand transfer products (*Au et al., 2004*; *Mizuuchi et al., 2007*). Both methods result in an off-pathway pseudo-disintegration reaction that has been referred to as a 'foldback' (see *Figure 5—figure supplement 1*). The foldback products imply that some of the STCs in those experiments were either destabilized prior to the assay itself or not properly assembled to being with. To avoid these issues, we devised a new protocol (*Figure 5—figure supplement 2*): STCs are formed by reaction of CDCs with labeled target DNA, but are immobilized on neutravidin-coated magnetic beads via biotinylated Mu end DNAs, such that remaining unreacted labeled target DNA can be rapidly washed away at the end of the STC formation step without stripping the $Mg^{2+}$ ions from the active site. Furthermore, an excess of high-affinity (mismatch-containing) cold competitor target DNA is added in the final wash step to trap any complexes that undergo subsequent disintegration. Any residual unreacted labeled target DNA that was not removed in the washing steps can be quantified by sampling the mixture immediately after purification. Monitoring these purified STCs as a function of time shows that they are impressively robust against disintegration under our normal reaction conditions, with, at most, less than 2% of strand transfer products reverting to re-ligated target DNA after 60 min (*Figure 5A*). This is true whether or not DMSO, a mismatch in the target DNA, or domain III were present. Overall, these data suggest that disintegration events themselves are normally rare, regardless of target binding affinity.

Previous studies have indicated several factors that could stimulate reversal: increased temperature, increased pH, and higher concentrations of glycerol in the reaction buffer (*Au et al., 2004*; *Lemberg et al., 2007*). We note that the first two modifications might weaken binding between MuA and bent target DNA. Indeed, we found that the combination of slightly increased buffer pH (7.9 instead of 7.4), increased glycerol content (16% instead of 5%), and high temperatures (60°C) triggered disintegration (*Figure 5B*). Under these conditions, WT STCs converted between 10–15% of strand transfer products back into intact target DNA after 1 hr, (*Figure 5C*). In our assay, this occurs without producing the 'foldback' products that have complicated previous studies (*Figure 5—figure supplement 1D,E*). Unlike the forward strand transfer reaction, disintegration was not highly affected by including a mismatch in the target DNA. Remarkably, when subjected to the same buffer and heat challenge, Δ domain III STCs catalyze 4-fold more disintegration than WT STCs, reverting about 40% of strand transfer products back into intact target DNA.

## Target DNA bending occurs upon binding but is deficient in Δ domain III CDCs

To observe more directly when the transpososome bends the target DNA and if the bend changes as a result of strand transfer, we measured Förster resonance energy transfer (FRET) between Atto565 and Atto647N fluorescent labels positioned at opposite 5' ends of our 35 bp target DNA. Target DNA bending would bring the labels at each end of the target DNA into closer proximity, reducing the fluorescence intensity and lifetime of the donor Atto565 label. We used time correlated single photon counting (TCSPC) to capture this decrease in donor lifetime. By measuring changes in

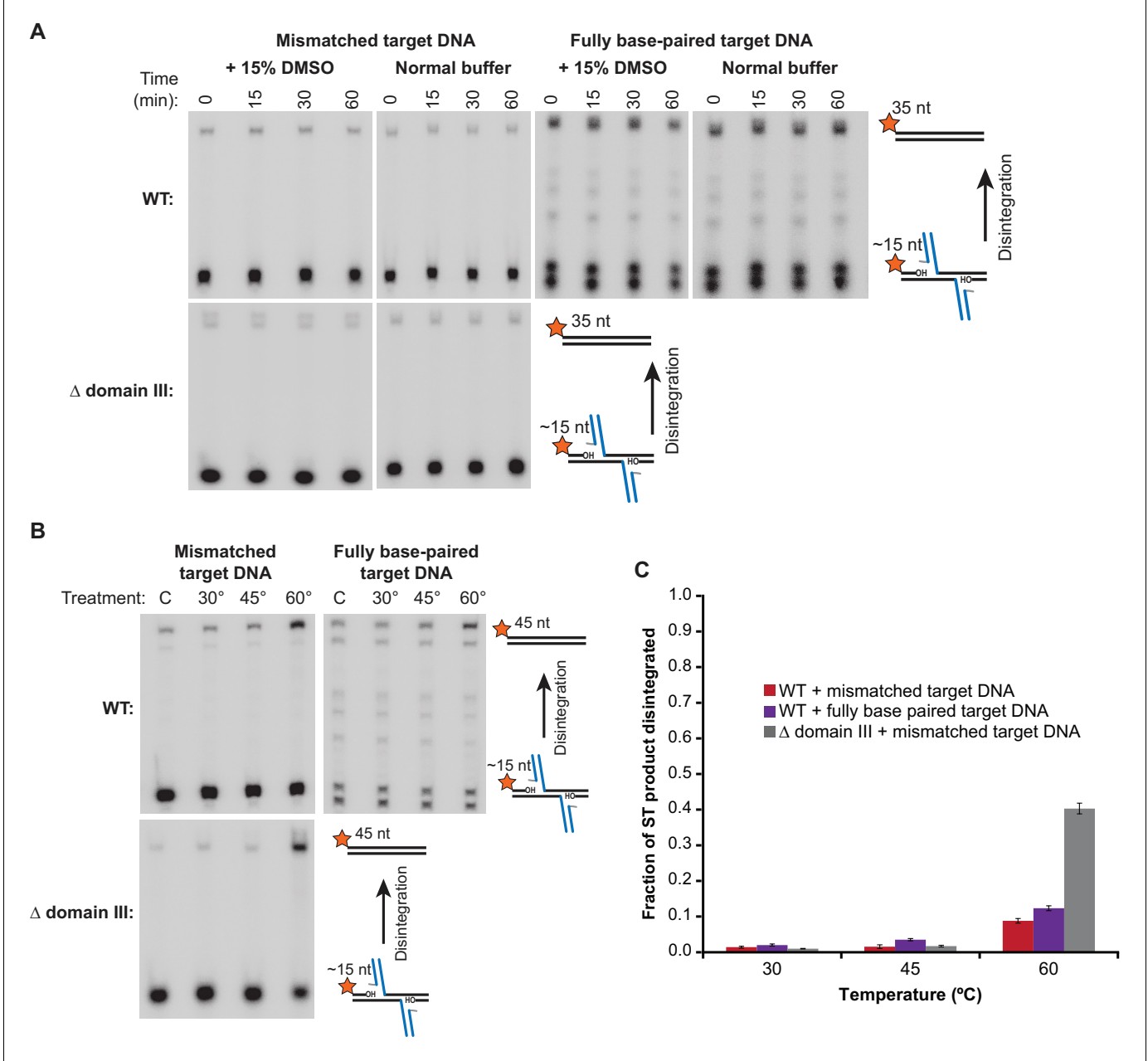

**Figure 5.** Strand Transfer Complex Disintegration. (**A**) Disintegration is not detectable under normal reaction conditions. Substrates and potential products were visualized by denaturing gel electrophoresis and 5'-$^{32}$P-labeled target DNA. STCs were rapidly purified by immobilization on magnetic beads into normal reaction buffer (see Materials and methods), which included DMSO where indicated, and incubated at 30°C. (**B**) Disintegration under modified reaction conditions. The same procedure as in (**A**), but STCs were purified into a modified buffer (see text) and held for 1 hr at the indicated temperatures. Target DNAs in these experiments have 5 bp added to each end to prevent melting during heating. Lanes labeled C are a sample taken immediately after purification, as in the 0 min timepoint in (**A**). (**C**) Quantification of replicates of the experiment shown and described in (**B**). The vertical axis represents the fraction of strand transfer product present immediately after purification that became re-ligated after treatment. Error bars represent mean ± SEM, n = 4 independent STC purifications.

The following figure supplements are available for figure 5:

**Figure supplement 1.** Pseudo-disintegration of strand transfer products by the 'foldback' pathway.

**Figure supplement 2.** Method for generating STCs for disintegration experiments.

fluorescence lifetimes, we are able to separate out contributions from unbound/unreacted target DNA and approximate the distribution of lifetimes (and thus the distribution of relative proximities) present in a sample (*Figure 6* and *Figure 6—figure supplement 1*). We do this by first observing the behavior of the doubly-labeled target DNA alone, free of any transpososomes (*Figure 6*, first column). The 35 bp target DNA should keep the fluorophore pair >116 Å apart, well outside the 69 Å Forster radius ($R_0$) for this dye pair. Indeed, the most reasonable fits to the decay of the DNA-only samples result from fitting a single discrete lifetime (shown as black bars) along with a minor (<10% of total photons) population of very short lifetimes (shown in grey). We take the former to be the lifetime of the donor fluorophore in unbent DNA (where FRET is negligible), and the latter to be a combination of imperfections in the instrumentation and anomalous fluorophore behavior. Decay measurements taken in the presence of transpososomes were then fit as the sum of this unbound fluorophore lifetime and a gaussian distribution representing any shorter lifetimes resulting from transpososome-induced bending.

The bend in bound target DNA prior to strand transfer can be measured by utilizing, once again, CDCs lacking the terminal Mu end 3'OH nucleophile for that reaction. Comparing normal conditions to either addition of DMSO or DMSO plus a mismatched base-pair in the target DNA, our analysis indicates that all three show a population of fluorophores in a bent (reduced fluorescence lifetime) configuration (*Figure 6*, second column). Once the contribution from unbound or unbent DNA is removed from each, the lifetime distributions from all three are centered at about 3.3 ns. This indicates that bound DNA is bent to approximately the same degree under all conditions and that enhanced flexibility acts largely to increase the bound/bent fraction. That the normal target DNA/ normal buffer condition shows a large unbound/unbent fraction is not surprising, given we have shown it exhibits poor affinity and struggles to utilize target DNA to completion. There is some suggestion in the fitted distributions that more flexible DNA is bent more stably, as judged by the tighter widths (FWHM) of the FRET distributions, but this trend may be outside of the resolution of this analysis. If strand transfer is allowed to proceed (by using unmodified Mu end DNAs; *Figure 6*, third column), the bend remains nearly unchanged for WT transpososomes across both the normal and highest flexibility conditions. Thus, any conformational changes in the target DNA brought about by strand transfer must be confined mainly to the vicinity of the nicks.

FRET also confirmed that Δ domain III CDCs have a reduced ability to bend target DNA. We have shown that they can bind relatively tightly to target DNA when both DMSO and a mismatched base pair are present (*Figure 3B*). However, FRET data for these same complexes indicates very little bending, despite the experiment being carried out at concentrations where the vast majority of the target DNA would be bound. This is direct evidence that our truncated CDCs are bending deficient. However, it is curious that despite a lack of bending they are still able to carry out strand transfer at a rate comparable to WT. It may be that there is a conformation of the target DNA that is competent for strand transfer (and still enhanced to a great degree by a base-pair mismatch) which does not necessarily bring the distant ends of the target DNA closer together, e.g. bends that are not properly phased for a U-turn, or it may be that the mismatch and DMSO allow rapid but very transient sampling of the strand-transfer-competent conformation. Nevertheless, after strand transfer proceeds and opens nicks in the target DNA, we find that the target DNA in Δ domain III STCs becomes significantly more bent and adopts a bend very similar to that seen in WT CDCs and STCs. This suggests that the breaks in the target DNA phosphate backbone allow it to access a bent conformation that is low enough in energy as to be resilient to the loss of some transpososome-DNA contacts. We also note that this convergence on the same bend coincides with the convergence in WT and Δ domain III behavior in resisting disintegration (at normal temperatures).

Finally, to determine whether unbending occurs under the high temperatures that trigger disintegration, we used a Peltier heater built into the cuvette holder of our TCSPC instrument to bring the mismatched target + DMSO STC complex samples rapidly to 60°C for 5 min prior to taking a repeated measurement. Heating the samples resulted in sizeable broadening of the FRET distribution, particularly for the disintegration-prone Δ domain III transpososomes, and an increase in the fitted amount of unbent target DNA (*Figure 6*, fourth column). This means that disintegration coincides with conditions where the target DNA conformation is less stably controlled (and a fraction of STCs may have lost the bend in the target DNA entirely).

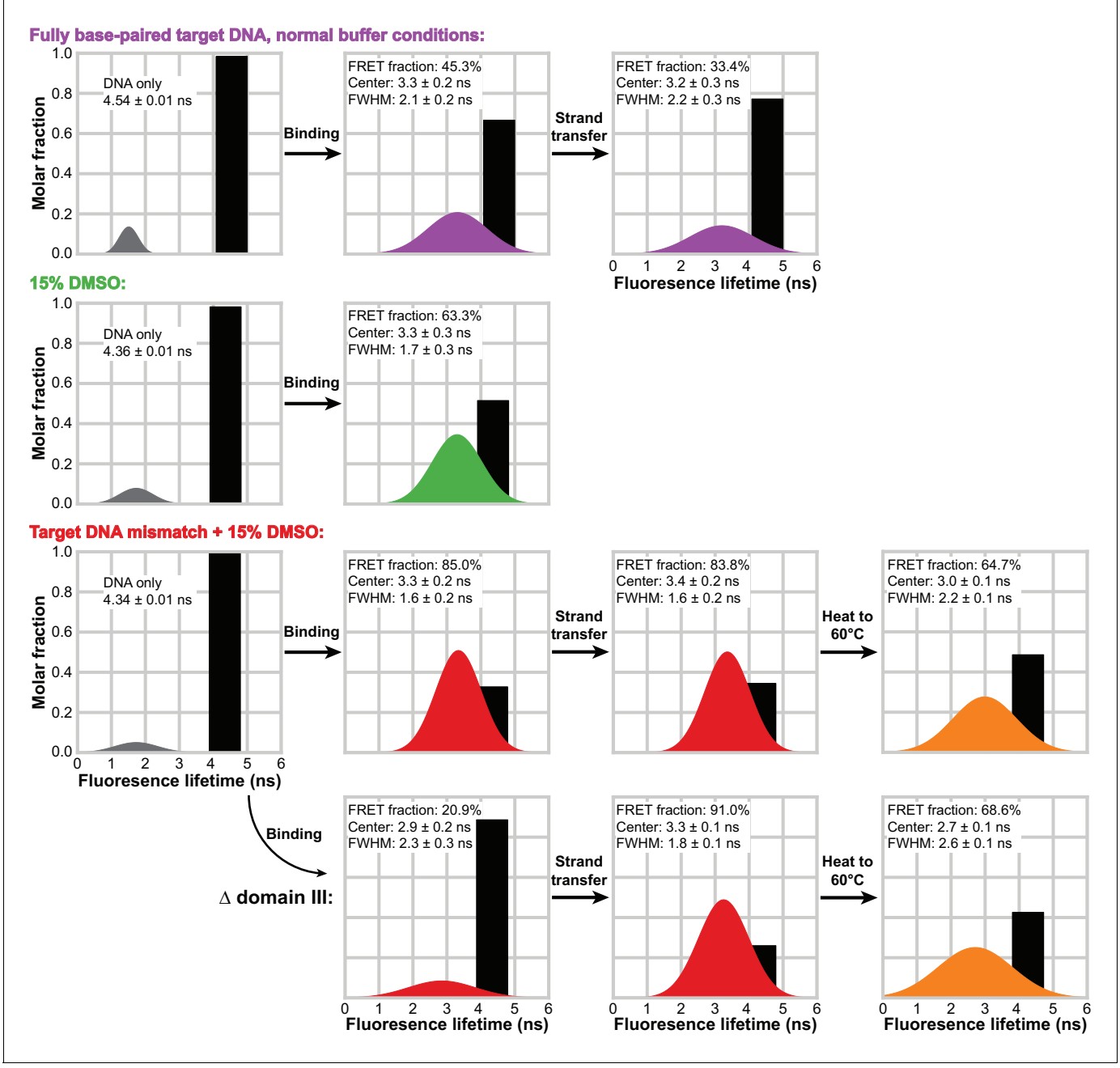

**Figure 6.** Target DNA bending measured by TCSPC FRET. The measured fluorescence decay for each condition was fit with a model combining a discrete lifetime (black bars) and a distribution of lifetimes (colored gaussian distributions). In each plot, the area under each component represents its molar fraction in the fit; the center, full width at half maximum (FWHM) and relative abundance is given for each FRET distribution. In the absence of transpososomes (left-most column), little to no FRET can be detected and the vast majority of fluorophores decay with a discrete lifetime. This discrete lifetime was held constant as CDCs lacking the strand transfer nucleophile (second column) and active CDCs (third column) were added. Δ domain III transpososomes were used for the bottom row. In the fourth column, the STC samples from the bottom two rows were heated to 60°C for 5 min and measured again.

The following source data and figure supplement are available for figure 6:

**Source data 1.** Time correlated single photon counting FRET data.
**Figure supplement 1.** Fluorescence lifetime fits across all samples.

## Discussion

In this study, we have shown that the conformation of the target DNA has dramatic consequences for transposition by phage Mu. Our model for this process is shown as a free energy reaction coordinate diagram in *Figure 7A*. In the absence of outside factors, newly assembled transpososomes bind to potential target DNA with low affinity. We hypothesize that this is because the target DNA must be strongly bent to optimize its contacts with CDCs, but free energy released by making those contacts does not compensate well for the conformational strain required to bend the target DNA. This is supported by our findings that enhancing the flexibility of the target DNA and pre-bending it both greatly enhance its affinity for CDCs, while removing some of the contacts that stabilize the bent form by deleting domain III greatly reduces its affinity. We also found that strand transfer rates correspond relatively well with target binding affinity, which suggests that getting target DNA into the active sites of the CDCs can be rate-limiting for strand transfer and thus provides a control point for determining the choice of target. This aspect of Mu transposition was exploited by *Yanagihara and Mizuuchi (2002)* who showed it can be used in vitro to pinpoint a single mismatch in the presence of a vast excess of unperturbed DNA. Furthermore, in vivo transposition into unperturbed target DNA may be even slower than in our assays, as the presence of intact flanking DNA has been shown to further inhibit target capture (*Williams and Baker, 2004*).

Why should transposition be slow by default? Why scan through potential target sites? Although Mu and many of its transposon and retrovirus relatives do not hunt for specific target sequences, they do need to avoid inserting into their own genomes even though their own DNA is necessarily present at high local concentration. We propose that transposition is repressed by default except in the presence of DNA bound by target selection proteins. For Mu, target selection is driven by the distribution of MuB protein, whose reported biochemical activities appear at first glance paradoxical (reviewed in *Dramićanin and Ramón-Maiques, 2013*). MuB forms filaments along dsDNA in the presence of ATP (*Maxwell et al., 1987*; *Mizuno et al., 2013*), and binds the C-terminal region of MuA (*Wu and Chaconas, 1994*). Contact with MuA, even the monomeric form present before a transpososome is formed, triggers ATP hydrolysis and DNA release by MuB (*Adzuma and Mizuuchi, 1988*; *Greene and Mizuuchi, 2002*). Thus DNA near the Mu ends is probably cleared of MuB before an active transpososome is formed. However, MuB-bound DNA molecules are strongly preferred as transposition targets. Until recently, it has been difficult to explain how DNA sites coated or encased in MuB can be attractive targets. Our results suggest how transposition targeting mechanisms might activate transposition: their ability to present pre-deformed or pre-bent DNA. The structural data available for MuB and IstB (the related targeting ATPase from IS21elements) suggest that they have the ability change the helical conformation of the DNA they coat and/or to form bundles of protein-DNA filaments (*Arias-Palomo and Berger, 2015*; *Dramićanin et al., 2015*; *Mizuno et al., 2013*). Loops could be extruded from these large protein-DNA complexes by transient dissociation events within a filament or as the DNA crosses from one collinear filament to an adjacent one within a bundle. These loops could act like our pre-bent minicircles, triggering tight binding and rapid transposition. This is supported by the observation that Mu has a bias towards target sites immediately adjacent to MuB-coated DNA, rather than inside MuB filaments themselves (*Ge et al., 2011*). For retroviral integration, nucleosomes provide pre-bent DNA by their very nature. Retroviral integration happens directly on the nucleosome, and a high-resolution structure of this event has been obtained by cryo-electron microscopy (*Maskell et al., 2015*). Nucleosome DNA could be a favored target as long as the energy needed to dislodge a DNA loop slightly from the nucleosome surface and into the integrase active sites is less than that required to bend DNA de novo. There is biochemical evidence for this energetic compromise, as integration is known to occur predominantly at positions that are bound least tightly to the histone core (*Maskell et al., 2015*; *Serrao et al., 2015*).

Much of the conformational strain accrued by the bent target DNA could be released by strand transfer nicking the target DNA backbone. These nicks would allow the adjacent DNA to locally relax to a less strained conformation while maintaining or strengthening the energetically favorable MuA-target contacts. Additionally, the structural consequence of relaxing the strain in the bent target DNA could be to eject one or both of the strand transfer product moeities (the target 3'OH or new phosphodiester bond) from the active site, which is the case for the four available strand transfer complex structures (*Maertens et al., 2010*; *Montaño et al., 2012*; *Morris et al., 2016*; *Yin et al., 2016*). The equilibrium of the strand transfer reaction may therefore be strongly biased toward

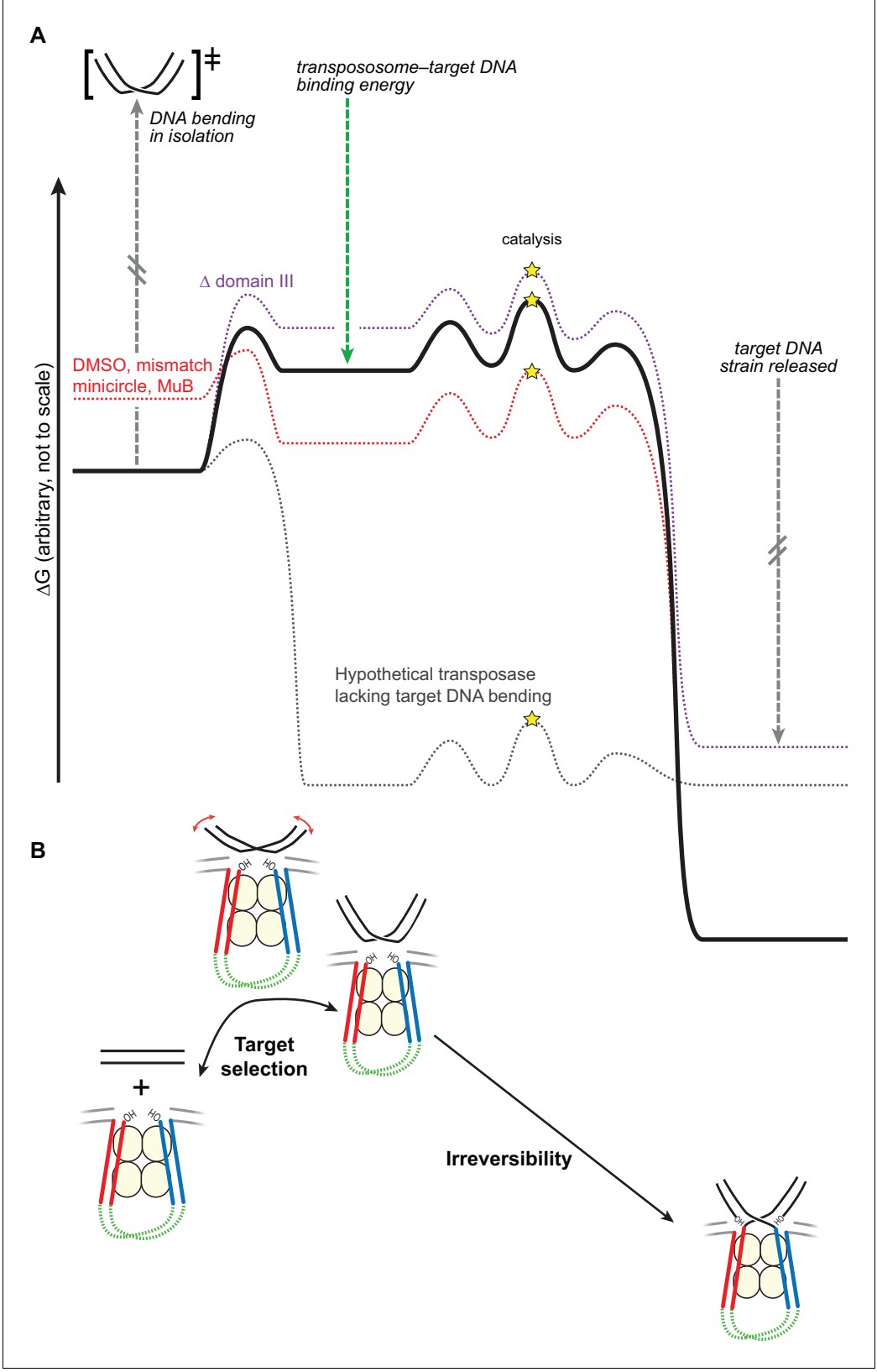

**Figure 7.** Bending the target DNA slows target capture and resists disintegration. (**A**) Cartooned as a simplified free energy diagram, with the unmodified system as the bold black line. Target DNA capture is energetically uphill because the target DNA must be bent. This equilibrium can be altered to favor target capture by factors that

*Figure 7 continued on next page*

*Figure 7 continued*

promote DNA flexibility (here, mismatched base-pairs, DMSO buffer, or mini-circularization, dashed red line), or to make target capture even more uphill by removing some contacts between the transpososome and the target DNA (here, deleting domain III, dashed purple line). Strand transfer interrupts the target DNA backbone, thus releasing the conformational strain induced by the bend while preserving or solidifying the protein-DNA contacts between the transpososome and target. This leaves the strand transfer products in a deep thermodynamic optimum that prevents disintegration. Our results indicate that domain III stabilizes the target DNA conformation to a greater degree after strand transfer. The dashed grey line represents a hypothetical transposase that binds to target DNA favorably and without inducing any conformational strain. (**B**) The features of the energy landscape outlined in (**A**) have the effect of resisting target DNA capture until an outside factor (e.g., MuB) can compensate for the uphill free energy difference. After strand transfer, the release of bending strain combined with product binding energy ensure that strand transfer is effectively irreversible.

products because the final strand transfer complex is much lower in energy than the initial states (CDCs both before and after capturing target DNA).

This thermodynamic minimum may be so deep that thermal energy is insufficient to bring the 3' OH and scissile phosphate group back into reacting position at any appreciable rate. The existence of a kinetic barrier to disintegration is supported by our finding a lack of disintegration products at 30 degrees, even under conditions where we would have expected any transiently-formed disintegrated target DNA dissociate because its affinity for CDCs is so weak: $\Delta$ domain III transpososomes with no DMSO, and fully base paired DNA with WT transpososomes and no DMSO (and with an excess of higher affinity cold target DNA present to trap any released CDCs). This implies that under these conditions, the transpososomes simply do not sample the equilibrium between the target capture and strand transfer complexes.

Disintegration could be detected, however, at high temperature, which FRET showed also destabilizes the conformation of the STC target DNA. We suggest that heat weakens the protein-DNA contacts holding the strand transfer product moieties away from the active site and provides the thermal energy necessary to escape the STC energy minimum. Disintegration at 60 degrees was much more efficient for $\Delta$ domain III STCs than for WT ones. Because the $\Delta$ domain III CDCs have very low affinity for target DNA, this might be explained by their releasing transiently disintegrated DNA more readily. However, WT complexes have a much lower affinity for normal vs. mismatched target DNA, but these produced similar amounts of disintegration product. This suggests instead that domain III – target DNA interactions have an even stronger stabilizing effect on the final strand transfer complex than they do on the intermediate captured target, probably because the binding surface is particularly tailored to a target DNA conformation that is only readily accessible after nicking. Therefore removal of domain III appears to adjust the equilibrium between the target capture complex and the final STC.

An alternate way to consider the energetics of this reaction is to consider the free energy diagram if there were no DNA bending, or in the most extreme case, no contacts to any part of the DNA except the scissile phosphate. In that case (dashed grey line in *Figure 7A*), transpososomes would likely bind to any free DNA immediately as a target and there were be no energetic difference between the target capture complex and the final STC (the equilibrium while bound to the enzyme would be 1). Regarding the strand transfer step, we propose that removing domain III, heat, and the other buffer modifications that helped to promote reversal, move the system slightly toward this scenario by weakening the contacts between the target DNA and MuA. Heat also by definition helps cross the kinetic barrier between the two states (in both directions).

In summary, we find that target DNA is strongly bent upon initial capture by CDCs (as shown by the FRET data in *Figure 6*). Because the energy required to induce this bend in the DNA is only partially compensated for by energetically favorable protein-DNA contacts, CDCs have very weak affinity for unperturbed target DNA, which is reflected in very slow rates of strand transfer into those targets. As cartooned in *Figure 7A*, we found that this barrier can be overcome in several ways: by lowering the free energy of the bent DNA conformation, using DMSO or a mismatch, or by raising the free energy of the initial unbound state by ligating it into a strained pre-bent minicircle. We suggest that in the natural pathway for Mu transposition, MuB protein does the latter, presenting pre-

bent DNA to the CDCs. The affinity for target DNA could be manipulated in the opposite direction by removing domain III. That Δ domain III transpososomes are specifically defective in stabilizing DNA bending was supported by our observation that they could be rescued by increasing the flexibility of the target DNA or by pre-bending it. The strand transfer reaction greatly increases the stability of the final protein-DNA complex by releasing strain in the target DNA while maintaining, and probably increasing, the protein-DNA contacts. The stability of this complex shifts the equilibrium of the strand transfer reaction strongly towards products.

We therefore propose that target DNA bending therefore serves two purposes (*Figure 7B*): (1) it helps enforce proper choice by rendering binding of CDCs to naked target DNA very weak, and (2) it alters the energetic landscape in a way that drives the strand transfer reaction forward.

## Materials and methods

### Expression and purification of MuA and SinMu proteins

Proteins were expressed in the Rosetta DE3 Escherichia Coli strain (Merck KGaA, Darmstadt Germany) from coding sequences cloned into the pET3c plasmid. Transformed cells were grown at 37°C in LB media supplemented with 100 μg/mL ampicillin to OD600 ≈ 0.7, then protein expression was induced by addition of 0.66 mM IPTG and an additional 20 μg/mL ampicillin. At 2 hr after induction, the cells were collected by centrifugation at 8000 rpm, and stored at −80C until lysis. Cells were resuspended in a solution of 25 mM HEPES pH 7.5, 1 mM EDTA, 1 M NaCl, 10% sucrose, 10% glycerol, 5 mM DTT, and Complete protease inhibitor (Roche Diagnostics, Indianapolis IN) and lysed by two passes through an LV1 Microfluidizer (Microfluidics, Westwood MA). Cell debris was removed by centrifugation and the resulting supernatant was fractionated by addition of ammonium sulfate to 30% saturation. The precipitate was collected by centrifugation at 18,000 rpm in an SS-34 (Thermo Fisher Scientific, Waltham MA) rotor for 30 min, and the resulting pellet was resuspended in 20 mM MES pH 5.5, 0.2 M NaCl, 0.5 mM EDTA, 5% glycerol, and 1 mM DTT (Buffer A). This was passed over a HiPrep Heparin FF 16/10 affinity column (GE Healthcare, Chicago IL) and eluted with a gradient from 0.2 M (Buffer A) to 2 M (Buffer B) NaCl. Fractions containing the protein were diluted back into Buffer A and the heparin affinity chromatography was repeated. Fractions containing the protein were concentrated and further purified by gel filtration chromatography using HiLoad 16/600 Superdex 75 prep-grade column (GE Healthcare) equilibrated with 25 mM HEPES pH 7.5, 0.4 M NaCl, 0.5 mM EDTA, 5% glycerol, and 1 mM DTT. Peak fractions were combined, dialyzed into the same buffer supplemented to 20% glycerol, concentrated, and stored at −80°C. The final concentration of proteins was determined by measuring their absorbance at 280 nm.

### DNA substrates

The sequences of the linear DNA substrates used in this work are given in *Figure 1—figure supplement 1*. All oligonucleotides, including biotin and fluorophore modifications, were synthesized by Integrated DNA Technologies (IDT, Coralville IA). Oligonucleotides modified with biotin or fluorophores were HPLC purified by IDT, and $^{32}$P labeled oligonucleotides were PAGE purified by IDT. Oligonucleotides were resuspended in TE (10 mM Tris pH 8, 1 mM EDTA), and those not purified by IDT were desalted using P6 spin columns (Bio-Rad Laboratories, Hercules CA) equilibrated in TE. The concentration of all oligonucleotides was verified by measuring their absorbance at 260 nm. DNA substrates were radiolabeled at the 5' end using γ-$^{32}$P ATP (PerkinElmer, Waltham MA) and T4 polynucleotide kinase (Thermo Fisher Scientific). Oligonucleotides were 3' modified with dideoxyadenosine by 4 hr treatment with Terminal Transferase (New England Biolabs, Ipswich MA) and a 25-fold molar excess of dideoxy-ATP. Double stranded DNA substrates were created by mixing complementary oligonucleotides of the desired sequence in equimolar amounts in a buffer of 10 mM Tris pH 8, 100 mM NaCl, and 1 mM EDTA, heating to 80°C, and then slowly cooling to room temperature.

### Transposition reactions

CDCs were formed in a buffer of 25 mM HEPES pH 7.4, 200 mM NaCl, 5% glycerol, 0.6 mM Zwittergent 3–12 (Merck KGaA), and 0.5 mM EDTA. The reaction buffer used during experiments was identical to this, except EDTA was omitted and replaced with 10 mM MgCl$_2$. Where indicated, these

buffers also contained 15% (v/v) DMSO. Unless otherwise specified, this reaction buffer was used for all experimental procedures involving transpososomes. Cleaved donor complexes (CDCs) were formed by mixing protein and Mu end DNAs in a 2:1 MuA protein:Mu end DNA ratio (or a 1:1:1 MuA protein:SinMu protein:Mu end DNA ratio) and incubating at 30°C for $\geq$1 hr. In cases where gel filtration was used to purify CDCs, DMSO was included in the formation buffer. Purification of CDCs by gel filtration was performed using a Superdex 200 Increase 10/300 column (GE Healthcare) equilibrated in formation buffer lacking DMSO. After gel filtration the tetramer peak was collected, spin concentrated, quantified by measuring absorbance at 260 nm, and used immediately. Target binding or strand transfer reactions were carried out at 30°C and initiated by diluting fresh CDCs and the appropriate target DNA into reaction buffer. This buffer contained DMSO where indicated. For electrophoretic analysis, strand transfer reactions were stopped by phenol:chloroform extraction.

## Fluorescence anisotropy

CDCs were formed as described above, except with DNAs ending in dideoxy-A and in reaction buffer (with mM MgCl$_2$), which was present during all steps. Purified CDCs were serially diluted (in reaction buffer), mixed with 6 nM Atto565-labeled target DNA, and arrayed into a Corning 3575 black polystyrene 384 well microplate (Corning, Corning NY). Binding reactions were incubated for 1 hr at room temperature. A Victor X5 plate reader (PerkinElmer) was used to read the anisotropy of the Atto565 fluorophore, using a 531 nm excitation and 595 nm emission filters and a 1.0 s counting time. The raw parallel and perpendicular photon counts were adjusted to account for the instrument response (G factor and plate positioning artifacts), and the resulting anisotropies were fit to an equilibrium binding model, accounting for receptor depletion, using the optimize.curve_fit function from the Python scipy package (Research Resource Identifier (RRID:SCR_008058). For each data point, 15 total measurements of anisotropy were made: three separate serial dilutions each measured by the plate reader five times.

## Minicircle DNA

Minicircle DNAs were constructed by ligating together two linear DNA fragments with complementary sticky ends. These fragments contained a binding site for the Integration Host Factor (IHF) protein, which was added prior to ligation so that the DNA fragments would be bent in order to encourage circular ligation. IHF protein was purified using a protocol described previously (*Swinger and Rice, 2007*). The sequences of the minicircle DNAs used in this work are given in *Figure 4—figure supplement 2*. In detail, the two DNA pieces were mixed in an equimolar ratio along with a twofold molar excess of IHF in T4 Ligase buffer (New England Biolabs) for 30 min at room temperature. T4 DNA Ligase (New England Biolabs) was then added and the mixture incubated for 12 hr at room temperature. Proteins were removed by phenol:chloroform extraction followed by P6 column (Bio-Rad Laboratories) buffer exchange into fresh T4 Ligase buffer. T4 DNA Ligase was added back in and incubated at room temperature for an additional 2 hr to remove any remaining nicks. This reaction mixture was phenol:chloroform extracted and separated by gel electrophoresis on an 8% polyacrylamide TBE gel. The band corresponding to the circular product was identified by UV shadowing, excised, and extracted by shredding the gel slice and soaking in a ~15 fold volume excess of 10 mM Tris pH 8, 250 mM NaCl, and 2 mM EDTA for 8 hr at room temperature. Gel fragments were removed by filtration. To remove any remaining linear fragments or nicked circles, an equal volume of 2x BAL-31 nuclease buffer and 5 µL BAL-31 nuclease (New England Biolabs) were added and incubated for two hours each at 30°C and 40°C. The nuclease was removed by addition of EGTA to 20 mM followed by phenol:chloroform extraction and P6 column buffer exchange into a buffer of 10 mM Tris pH 7.5, 100 mM NaCl, 15% glycerol, 2 mM EDTA, and 2 mM EGTA. Minicircles were stored at −20°C.

## Disintegration

CDCs were formed as described above using Mu end DNAs modified with biotin on the 5' end of the transferred strand (*Figure 1—figure supplement 1*). After allowing an hour for CDC formation, Sera-Mag SpeedBeads NeutrAvidin particles (GE Healthcare) were added to ~0.1% (w/v) final concentration and incubated for an additional 30 min at 30°C. Using a magnet to immobilize bead-bound CDCs, the mixture was washed to remove unbound protein and DNA and replace the CDC

formation buffer with reaction buffer (including $MgCl_2$ and DMSO). An approximately two-fold molar excess of the indicated target DNA was then added for STC formation. This strand transfer reaction proceeded for 2 hr at 30°C, with gentle mixing every 30 min to keep the magnetic beads evenly dispersed. During the final 5 min, heparin was added to a final concentration of 0.05 mg $mL^{-1}$ to encourage dissolution of any incompletely formed MuA-DNA complexes. Immobilized strand transfer complexes were then washed four times into the desired buffer for disintegration (which always contained $MgCl_2$). A ~5 fold molar excess of unlabeled cold competitor mismatched target DNA was added in the final wash. A sample was always taken immediately after the final wash (time = 0 min) to record the initial baseline levels of reacted/unreacted DNA. After this, the reaction was either sampled as a function of time or divided equally among three temperature conditions (30°C, 45°C, and 60°C) for 1 hr. Disintegration was carried out where indicated in either the standard reaction buffer used throughout this work, or reaction buffer modified to pH 7.9% and 16% glycerol. Samples of the reaction were taken by dilution into a 10-fold volume excess of 97% formamide +10 mM EDTA that had been pre-heated to 100°C.

## Time correlated single photon counting FRET

Samples included 100 nM of labeled target DNA and either 1.5 µM (for the fully base-paired target DNA / no DMSO condition) or 1.0 µM (all other samples) purified CDCs. This mixture was equilibrated at 30°C for 1 hr prior to measurements. For each condition, measurements were made of both a donor-only (Atto565) and a donor-acceptor (Atto565 + Atto647N) sample. Time-domain lifetimes were measured on a ChronosBH lifetime fluorometer (ISS, Inc., Champaign IL) using Time-Correlated Single Photon Counting (TCSPC) methods. The fluorometer contained Becker-Hickl SPC-130 detection electronics and an HPM-100–40 Hybrid PMT detector (Becker & Hickl GmbH, Berlin Germany). Tunable picosecond pulsed excitation at 565 nm was provided by a Fianium SC400–2 supercontinuum laser source with integrated pulse picker and AOTF (NKT Photonics, Birkerød Denmark). Emission wavelengths were selected with a Semrock Brightline FF01-593/40 bandpass filter (Semrock, Inc., Rochester NY). The Instrument Response Function (IRF) was measured in a 1% scattering solution of Ludox LS colloidal silica in matched buffers. Multi-component exponential decay lifetimes and distributions were fit via a forward convolution method in the Vinci control and analysis software (ISS, Inc.). For both donor only and donor-acceptor samples, all fitted parameters are given in *Figure 6—figure supplement 1*. Raw data in both the original and a tab-separated plain text format, as well as fitting outputs from the Vinci software in PDF format, are available for download as *Figure 6—source data 1*.

Plots were prepared using the Python package Matplotlib (RRID:SCR_008624). Visualization and rendering of macromolecular structures for figures used Pymol (The PyMOL Molecular Graphics System, Version 1.8 Schrödinger, LLC; RRID: SCR_000305).

## Acknowledgements

James R Fuller received support from NIH training grant GM007183. We thank Ying Z Pigli for purified IHF protein and general laboratory assistance, and Sherwin P Montaño for implementing the SinMu system. The SinMu expression plasmid was a gift from Tania Baker (Massachusetts Institute of Technology). We thank Anjum Ansari and Viktoriya Zvoda (University of Illinois) and Justin E Jureller (University of Chicago) for advice regarding TCSPC experiments. We thank the lab of Joe Thornton (University of Chicago) for use of their fluorescence anisotropy plate reader, which was supported by NIH grant R01GM104397. TCSPC instrumentation was provided by the MRSEC Shared User Facilities (NSF DMR-1420709) and Institute for Biophysical Dynamics NanoBiology Facility Instrumentation (NIH 1S10RR026988-01) grants at the University of Chicago.

## Additional information

### Funding

| Funder | Grant reference number | Author |
|---|---|---|
| National Institute of General Medical Sciences | GM101989 | Phoebe A Rice |

The funders had no role in study design, data collection and interpretation, or the decision to submit the work for publication.

## Author contributions

JRF, Conceptualization, Data curation, Software, Formal analysis, Investigation, Visualization, Methodology, Writing—original draft, Writing—review and editing; PAR, Conceptualization, Resources, Supervision, Funding acquisition, Project administration, Writing—review and editing

## Author ORCIDs

James R Fuller, http://orcid.org/0000-0002-9029-0923
Phoebe A Rice, http://orcid.org/0000-0002-3467-341X

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
