## [Decision Letter]

Thank you for submitting your article "Target DNA bending by the Mu transpososome promotes careful transposition and prevents its reversal" for consideration by *eLife*. Your article has been favorably evaluated by Michael Marletta (Senior Editor) and three reviewers, one of whom is a member of our Board of Reviewing Editors. The reviewers have opted to remain anonymous.

The reviewers have discussed the reviews with one another and the Reviewing Editor has drafted this decision to help you prepare a revised submission.

Summary:

The three reviewers found your manuscript entitled 'Target DNA bending by the Mu transpososome promotes careful transposition and prevents its reversal' an interesting and, for the most part, well-written manuscript addressing a central question in the regulation of DNA transposition. The paper shows that bending of the target DNA is a key regulatory step where they isolate target binding and cleavage from the earlier processing steps using an array of biochemical and FRET-based methods. Furthermore, it also presents a convincing model explaining why the Strand Transfer Complex (STC) is so stable that its disassembly requires the action of an ATPase.

Essential revisions:

Overall the conclusion was that the manuscript would benefit from some revision to increase its general accessibility, and to resolve some issues of clarity and mechanism. Below I list the issues that in our opinion need addressing before a decision on publication can be made.

1) Text and numbers in some of the Figures (e.g. Figure 2, Figure 3, Figure 4, Figure 5) are hard to read and the figures are very crowded. Furthermore, there is a range of font sizes and font styles-regular vs. bold. In the second paragraph of the subsection “MuA domain III participates in target DNA binding and bending”, 'select conditions' does not seem right.

2) Introduction, second paragraph: "Host-transposon boundary" is inappropriate. Please re-word (e.g., 'cleavage at specific sites at the transposon ends'. In the in vitro experiments discussed in this paper, there is no 'host'.

3) Subsection “Flexible or bent DNA is a highly reactive transposition target”, first paragraph: Sentence has been truncated?

4) Section "Strand transfer reversal only occurs under conditions that would compromise target DNA binding…": This section should be substantially rewritten, as we did not find it particularly straightforward to understand, especially for someone who is not familiar with Mu transposition. It would be important to clearly define the two possible assembly methods of the STC and explain why one particular method was chosen here. Please define "excess components". Please find some way more conducive to the reproducibility of these experiments to describe STC purification more precisely than "rapidly and gently" (subsection “Strand transfer reversal only occurs under conditions that would compromise target DNA binding”, first paragraph). I think the lack of a clear description of the different possible assembly methods also shows up at "unreacted labeled target DNA" Unreacted at which step? I think it would also be helpful to include a panel in Figure 5 showing exactly the steps in these experiments.

In the second paragraph of the aforementioned subsection: I think it would also be useful to add a few lines pointing out that in the X-ray structure described by Montano et al., the STC was also assembled (like here) from components of Mu ends and target that they reacted after assembly. Please correct us if we are wrong.

5) A second issue related to target bending/flexibility is the observation that reversal of the strand transfer reaction is normally a rare event. Given that the reaction is overall isoenergetic and disintegration is unimolecular, the mechanistic basis for this has not been clear. We found the authors' discussion of this phenomenon in terms of their experimental results (Discussion, last paragraph) to be difficult to follow. The idea of "a deep energetic well out of which the strand transfer products rarely reverse, despite the otherwise enthalpically neutral (and likely entropically uphill) nature of that chemical reaction" seems to be a thermodynamic argument that the STC product of the reaction has a lower free energy than the pre-cleavage complex. In this scenario, disintegration is energetically uphill because the complex needs to reach a pre-cleavage like configuration in order to reverse the reaction. Wouldn't this also imply that bending-compromised complexes would have even more difficulty reversing the reaction since they would be going even more uphill? The other point of confusion here is that the lack of reversal could be primarily a kinetic barrier; this could be what the authors had in mind, but if so it is not clear.

Relating to this, one reviewer was confused by Figure 7. The figure indicates that the target capture complex (prior to cleavage) has a higher free energy than the reactants, yet the authors demonstrate Kd values of 15-30 nM for this binding event under bending-favored conditions (i.e., std deltaG << 0). It is therefore hard to reconcile the rest of the figure. It seems that one could explain disintegration rates based on the stability of the target capture complex alone. For a stable complex (DMSO, mismatch), disintegration could occur at some rate, but the stability of the disintegration product just promotes target cleavage again and the energy relaxation resulting from nicking (as suggested by the authors) favors the STC. For unstable complexes, disintegration will result in a greater dissociation, giving the net effects observed.

6) It would be helpful if the authors included a figure similar in content to that of Figure 2, Montano et al. Nature 491, page 414. This is because as it is now, Figure 1 doesn't allow a clear understanding of either the significance of domain III or that of the SinMu chimera. Given that the comparison with the other target-containing transposomes is important; this additional figure should be a new panel somewhere. The best place to reference such a figure (or panel) would probably be somewhere in the first paragraph of the subsection “MuA domain III participates in target DNA binding and bending”.

7) In the Introduction, the passage “Transposases and integrases face two challenges when interacting with target DNA sites…”: the authors should consider rewriting this, as we are not aware that target immunity has been shown to be a general feature of transposition. In fact, there may be more DNA transposons that do not care about inserting into themselves.

8) Introduction, third paragraph: along these lines, please replace "many" with "some".

9) Subsection “Flexible or bent DNA is a highly reactive transposition target”, fifth paragraph, and in other instances when "transpososomes" are mentioned: I think, for clarity, it would be very useful to specify whether we are talking about the CDC or the STC in all cases through the manuscript.

10) Subsection “MuA domain III participates in target DNA binding and bending”, second paragraph: "virtually required" Please rephrase; what is a 'virtual requirement'?

11) Discussion, second paragraph: "We propose…". I think this sentence is perhaps the most important conclusion in the work, and the arguments that follow put the in vitro data in the manuscript into a biological context. I think that more effort should be spent shedding light onto the seeming paradox: MuB-coated DNA mediates immunity, but in the same time it delivers target by pre-bending it. I do not believe that the Discussion, as written now, is sufficient to explain to the uninitiated the relationships between concepts such as "avoid self-insertion", "MuB filaments" (why are these relevant?) and the role of MuB in target delivery.

12) Figure 1: Show that the second step is reversible (with arrow) and annotate as disintegration.

13)Subsection “Flexible or bent DNA is a highly reactive transposition target”, first paragraph: Not very clear; please reword or add the idea of "pre-cleaved transposon ends" for general readers.

14) The N-terminal domain that is left out is assumed to not be relevant for the binding/bending questions being addressed. What is the evidence for this assumption?

15) The G-G mismatch-containing targets used were based on earlier work in the Mu field. Do small bulges work in this system as well? That would seem to be an ideal way to direct and stabilize a bend.

16) Figure 2: The experiments shown here do not distinguish between single and double target cleavage events. The authors should identify experiments where this is the case and discuss whether it impacts the conclusions.

17) Figure 5: Nothing is happening as a function of time in this experiment. What is the point?

18) It might be worth pointing out that the overall Mu reaction does ultimately get driven by input of external energy. The stability of the initial complexes formed comes with the cost of needing to hydrolyze ATP (by ClpXP) to take the thing apart at the end. Not tightly coupled, but perhaps worth noting in the context of energetic arguments about the course of the overall reaction.

---

## [Author Response]

*Essential revisions:*

*Overall the conclusion was that the manuscript would benefit from some revision to increase its general accessibility, and to resolve some issues of clarity and mechanism. Below I list the issues that in our opinion need addressing before a decision on publication can be made.*

1) Text and numbers in some of the Figures (e.g. Figure 2, Figure 3, Figure 4, Figure 5) are hard to read and the figures are very crowded. Furthermore, there is a range of font sizes and font styles-regular vs. bold.

We have made small positional changes in the layout of panels in these Figures to alleviate crowding, although they were designed to visually summarize a large quantity of information.

Throughout the figures, font size is standardized to 8 point Arial in all cases except in which this would make secondary text too large to fulfill its purpose (e.g., size labels for DNA ladders or gel lane labels), in which case the font size is 7 point. For graphs, axis labels are bolded; similarly, for gel images, conditions that are consistent across many lanes or the entire sub-image are bolded. All other text has no style applied. This is broadly consistent with what we see in other articles published in *eLife*. That being said, we are happy to accommodate any additional, specific stylistic suggestions for improving the clarity of our figures.

It is also possible that the conversion to raster.png format, embedding in Word, and subsequent PDF conversion for preliminary submission may have degraded image quality. We expect that the final product generated by *eLife* from our full-resolution vector graphics files will appear easier on the eyes.

In the second paragraph of the subsection “MuA domain III participates in target DNA binding and bending”, 'select conditions' does not seem right.

We have replaced “select” with “particular” to avoid confusion.

*2) Introduction, second paragraph: "Host-transposon boundary" is inappropriate. Please re-word (e.g., 'cleavage at specific sites at the transposon ends'. In the* in vitro *experiments discussed in this paper, there is no 'host'.*

We have reworded this to clarify exactly where the cleavage occurs, and have made sure that we refer to this segment of DNA in all our experiments as “flank” rather than “flanking host”. However, we would like to retain the reference to “host” DNA in this introductory paragraph, as here we are discussing the full in vivo transposition process of phage Mu.

In response to this and other reviewer comments, Figure 1 has been modified to further emphasize that it refers to the full in vivo system by the addition of a cartoon of an entire host genome. In addition, the fact that our in vitro components are pre-cleaved only have 3 nt of flanking DNA is now more clearly stated in the opening paragraph of the Results section.

*3) Subsection “Flexible or bent DNA is a highly reactive transposition target”, first paragraph: Sentence has been truncated?*

Correct, we apologize for this error. The fix helps address part 13 below.

*4) Section "Strand transfer reversal only occurs under conditions that would compromise target DNA binding": This section should be substantially rewritten, as we did not find it particularly straightforward to understand, especially for someone who is not familiar with Mu transposition. It would be important to clearly define the two possible assembly methods of the STC and explain why one particular method was chosen here. Please define "excess components". Please find some way more conducive to the reproducibility of these experiments to describe STC purification more precisely than "rapidly and gently" (subsection “Strand transfer reversal only occurs under conditions that would compromise target DNA binding”, first paragraph). I think the lack of a clear description of the different possible assembly methods also shows up at "unreacted labeled target DNA" Unreacted at which step? I think it would also be helpful to include a panel in Figure 5 showing exactly the steps in these experiments.*

We apologize for the confusion in this section. We have rewritten the entire section for clarity, and added a cartoon of our method (Figure 5—figure supplement 2). We hope it is now clearer that here we *avoid* the STC purification methods used in the two previously published works because they generate the problematic “foldback” side product. We find that using magnetic beads to purify the STC (which, to our knowledge, is an approach to this problem that had previously not been tried) retains their activity and avoids foldback while allowing them to be purified in their normal buffer solution in a matter of minutes – hence “rapidly and gently”. Nevertheless, we have removed those adverbs from the text to avoid confusion.

*In the second paragraph of the aforementioned subsection: I think it would also be useful to add a few lines pointing out that in the X-ray structure described by Montano et al., the STC was also assembled (like here) from components of Mu ends and target that they reacted after assembly. Please correct us if we are wrong.*

We have added a note about how the STCs for crystallization were assembled (it was similar to that used by the Harshey group [Au et al., 2004], but without the gel purification) (the added reference can be found in subsection “Strand transfer reversal only occurs under conditions that would compromise target DNA 234 binding”, second paragraph). Here we did assemble STCs from components that reacted in situ rather than from branched “product” DNAs, but in this work we preformed CDCs on magnetic beads and washed away excess MuA and Mu end DNAs before adding target DNA and Mg^2+^. This separation of the assembly of CDCs from target DNA capture and strand transfer was not done for crystallization or by Au et al.

*5) A second issue related to target bending/flexibility is the observation that reversal of the strand transfer reaction is normally a rare event. Given that the reaction is overall isoenergetic and disintegration is unimolecular, the mechanistic basis for this has not been clear. We found the authors' discussion of this phenomenon in terms of their experimental results (Discussion, last paragraph) to be difficult to follow. The idea of "a deep energetic well out of which the strand transfer products rarely reverse, despite the otherwise enthalpically neutral (and likely entropically uphill) nature of that chemical reaction" seems to be a thermodynamic argument that the STC product of the reaction has a lower free energy than the pre-cleavage complex. In this scenario, disintegration is energetically uphill because the complex needs to reach a pre-cleavage like configuration in order to reverse the reaction. Wouldn't this also imply that bending-compromised complexes would have even more difficulty reversing the reaction since they would be going even more uphill? The other point of confusion here is that the lack of reversal could be primarily a kinetic barrier; this could be what the authors had in mind, but if so it is not clear.*

In retrospect, we agree that the way we framed the role of DNA bending was confusing. We have rewritten most of the Discussion and revised the cartoon in Figure 7 to clarify our ideas.

Briefly, we think that the equilibrium difference in the free energy of the intermediate (target capture) and final (strand transfer complexes) generates, via the “depth” of the strand transfer complex’s energy minimum, what is effectively (as the reviewer rightly points out) a kinetic barrier to disintegration (see Discussion, fourth paragraph). We believe our data support this sort of barrier, as opposed to a more “level” energy landcape where the CDC+target DNA and STC are of similar free energies but separated by a very high energy transition state that is unlikely to be crossed.

Similarly to the other reviewer comment addressing disintegration below, we too had considered that Δ domain III STCs might disintegrate at a rate comparable to the WT when heated, if domain III had a *uniform* negative impact on the stability throughout target catpure and after strand transfer. However the data showed that this was not the case. While domain III stabilizes both complexes, because Δ domain III STCs disintegrate at a higher rate it must be that domain III has an even greater stabilizing effect on the STC than during target capture by the CDC. We hope this is better reflected in the new Figure 7 and the new portion of the Discussion that describes it (see Discussion, fifth paragraph).

*Relating to this, one reviewer was confused by Figure 7. The figure indicates that the target capture complex (prior to cleavage) has a higher free energy than the reactants, yet the authors demonstrate Kd values of 15-30 nM for this binding event under bending-favored conditions (i.e., std deltaG << 0). It is therefore hard to reconcile the rest of the figure.*

The main black line in Figure 7 was meant to reflect the normal behavior of the system without our in vitro DNA flexibility modifications. We observe very poor binding and reaction rates under these conditions, hence the uphill nature of target DNA capture. The dashed purple line, which was labeled with our in vitro modifications, removes the uphill aspect for this very reason. We have now redrawn the entirety of Figure 7 to address other comments while hopefully emphasizing these features.

*It seems that one could explain disintegration rates based on the stability of the target capture complex alone. For a stable complex (DMSO, mismatch), disintegration could occur at some rate, but the stability of the disintegration product just promotes target cleavage again and the energy relaxation resulting from nicking (as suggested by the authors) favors the STC. For unstable complexes, disintegration will result in a greater dissociation, giving the net effects observed.*

Prior to conducting these experiments, we had considered this as a possible result of the reversal experiments (see also our answer to point #5 above). The data did not end up supporting this idea, however: (1) In Figure 5, under normal temperature conditions, we observe no disintegration even in the case of removing DMSO from Δ domain III transpososomes, which renders the affinity of those CDCs for target DNA so poor that it is undetectable. This is now noted in the last paragraph of the Results subsection “Strand transfer reversal only occurs under conditions that would compromise target DNA binding” and in the fourth paragraph of the rewritten Discussion section. (2) Even under high temperature conditions where we are able to reliably see disintegration events, within WT STCs there is very little difference between mismatched target DNA (which has high target capture affinity) and fully base paired target DNA (which has nearly undetectable target capture affinity). We hope that our rewritten Discussion clarifies these points (fifth paragraph).

*6) It would be helpful if the authors included a figure similar in content to that of Figure 2, Montano et al. Nature 491, page 414. This is because as it is now, Figure 1 doesn't allow a clear understanding of either the significance of domain III or that of the SinMu chimera. Given that the comparison with the other target-containing transposomes is important; this additional figure should be a new panel somewhere. The best place to reference such a figure (or panel) would probably be somewhere in the first paragraph of the subsection “MuA domain III participates in target DNA binding and bending”.*

We agree that the reader should be made better aware of the structural context of domain III. To this end, we have replaced Figure 4 panel A with an annotated zoomed view of the MuA STC crystal structure. What used to be panel A in that Figure now appears as a supplemental panel under Figure 4—figure supplement 1. As suggested, we reference this new panel when domain III is first mentioned in the Results subsection “MuA domain III participates in target DNA binding and bending”.

*7) In the Introduction, the passage “Transposases and integrases face two challenges when interacting with target DNA sites…”: the authors should consider rewriting this, as we are not aware that target immunity has been shown to be a general feature of transposition. In fact, there may be more DNA transposons that do not care about inserting into themselves.*

*8) Introduction, third paragraph: along these lines, please replace "many" with "some".*

Thank you for pointing this out; we have altered the language (Introduction, third paragraph) accordingly. This is in fact an interesting point (how do DDE transposons without target immunity resolve these detrimental products?), although outside the scope of our work here.

In addition, we have made a small addition to the preceding sentence to further clarify that avoiding self-insertion is required for *successful* transposition, rather than a pitfall that all DDE transposition systems have been shown to avoid. We hope that this makes it clearer that there is a considerable advantage to avoiding self-insertion while not making the claim that this has been shown in all Mu-like transposons.

*9) Subsection “Flexible or bent DNA is a highly reactive transposition target”, fifth paragraph, and in other instances when "transpososomes" are mentioned: I think, for clarity, it would be very useful to specify whether we are talking about the CDC or the STC in all cases through the manuscript.*

We agree that using the specific acronyms enhances clarity, and have made this change throughout the text. There are remaining instances where we use “transpososomes” to mean all forms of the complex more generally.

*10) Subsection “MuA domain III participates in target DNA binding and bending”, second paragraph: "virtually required" Please rephrase; what is a 'virtual requirement'?*

We have changed “virtually required” to “nearly essential” to clarify.

*11) Discussion, second paragraph: "We propose…". I think this sentence is perhaps the most important conclusion in the work, and the arguments that follow put the in vitro data in the manuscript into a biological context. I think that more effort should be spent shedding light onto the seeming paradox: MuB-coated DNA mediates immunity, but in the same time it delivers target by pre-bending it. I do not believe that the Discussion, as written now, is sufficient to explain to the uninitiated the relationships between concepts such as "avoid self-insertion", "MuB filaments" (why are these relevant?) and the role of MuB in target delivery.*

We have completely rewritten this portion of the discussion to better introduce what is known about MuB and to better explain our ideas of how it works.

*12) Figure 1: Show that the second step is reversible (with arrow) and annotate as disintegration.*

We have changed Figure 1 to include a small arrow flanked by question marks in order to mark where the disintegration reaction fits in the cartoon and prime a reader to consider the reversibility of this step.

*13) Subsection “Flexible or bent DNA is a highly reactive transposition target”, first paragraph: Not very clear; please reword or add the idea of "pre-cleaved transposon ends" for general readers.*

Those lines (subsection “Flexible or bent DNA is a highly reactive transposition target”, first paragraph) have been re-worded and (we hope) are now more broadly accessible.

*14) The N-terminal domain that is left out is assumed to not be relevant for the binding/bending questions being addressed. What is the evidence for this assumption?*

The role of domain Iα in Mu transposition has been extensively studied. Domain Iα is a site-specific DNA binding domain whose binding site is known: an “enhancer” region in the Mu genome where binding guides transpososome assembly; whereas (as we discuss in the Introduction section) Mu target selection has relatively little regard for sequence. Furthermore, in the crystal structure of the Mu STC it seems unlikely that domain Iα, at the N-terminus of MuA, would be physically capable of reaching the target DNA at all, as the closest approach of the N-terminus of domain Iβ to the target DNA is 55Å.

In the hopes of guiding the curious reader, we have added an additional citation to this section – Mizuuchi and Mizuuchi, 1989, (subsection “Flexible or bent DNA is a highly reactive transposition target”, first paragraph) – where the role of this domain was first characterized. For additional clarity, we have also made it explicit in the text that it is site-specific.

Finally, domain Iα is not conserved in related transposases such as those of Tn7 and IS21. We verified this by submitting their sequences to the RaptorX modeling server. RaptorX chooses the Mu transposase structure (which begins with domain Iβ) as the best template from the entire PDB for most of the Tn7 sequence except for its N-terminal ~60 residues, which it models on a template that is structurally unrelated to MuA’s domain Iα. For IS21 transposase, RaptorX predicts that the first 24 amino acids are poorly ordered and begins matching that sequence to MuA’s domain Iβ at about residue 25.

*15) The G-G mismatch-containing targets used were based on earlier work in the Mu field. Do small bulges work in this system as well? That would seem to be an ideal way to direct and stabilize a bend.*

Early on in the development of this project, attempts were made to explore the consequences of bulges. We had envisioned them as an alternative way to promote disintegration, with the extra nucleotides allowing the target 3’OH to easily access the active site even in a bent state. We found, however, that they were not proficient in directing the bend (like a mismatch) – instead they resulted in a difficult-to-interpret ladder of products indicating strand transfer at many positions. As a result, we did not pursue the idea further.

*16) Figure 2: The experiments shown here do not distinguish between single and double target cleavage events. The authors should identify experiments where this is the case and discuss whether it impacts the conclusions.*

We have now tested this using hairpinned target DNA substrates, and found that the vast majority of our strand transfer products (94-98%) reflect double target cleavage events. This new data is added as Figure 2—figure supplement 2.

*17) Figure 5: Nothing is happening as a function of time in this experiment. What is the point?*

We apologize for the confusion on this. Although Figure 5 is, in a sense, a “negative result,” we feel that it is surprising and notable enough to include as a main figure. First, as stated throughout the text, if protein-DNA product binding energy is discounted, disintegration would be expected to occur robustly rather than (as we find) not-at-all. Second, it addresses a discrepancy in the existing literature about the rate at which Mu STCs will catalyze this reaction. We hope that our significant edits to the first paragraph of the disintegration section (in response to reviewer points 4 and 5 above) now make it more clear why this negative result is important to highlight.

We have also changed the legend to Figure 5 to state the absence of disintegration explicitly.

*18) It might be worth pointing out that the overall Mu reaction does ultimately get driven by input of external energy. The stability of the initial complexes formed comes with the cost of needing to hydrolyze ATP (by ClpXP) to take the thing apart at the end. Not tightly coupled, but perhaps worth noting in the context of energetic arguments about the course of the overall reaction.*

We have edited the second paragraph of the Introduction to better explain this and have added a new reference to the Baker lab’s work on ClpX for the curious reader.